# Novel Immunoregulatory Functions of IL-18, an Accomplice of TGF-β1

**DOI:** 10.3390/cancers11010075

**Published:** 2019-01-11

**Authors:** Beatrice Casu, Alessandra Dondero, Stefano Regis, Fabio Caliendo, Andrea Petretto, Martina Bartolucci, Francesca Bellora, Cristina Bottino, Roberta Castriconi

**Affiliations:** 1Dipartimento di Medicina Sperimentale, University of Genova, 16132 Genova, Italy; beatricecasu90@gmail.com (B.C.); alessandra.dondero@unige.it (A.D.); fabiocaliendo2@gmail.com (F.C.); francesca.bellora@gmail.com (F.B.); roberta.castriconi@unige.it (R.C.); 2Istituto Giannina Gaslini, 16147 Genova, Italy; stefanoregis@gaslini.org (S.R.); a.petretto@gmail.com (A.P.); martinabartolucci@gaslini.org (M.B.); 3Centro di Eccellenza per la Ricerca Biomedica (CEBR), University of Genova, 16132 Genova, Italy

**Keywords:** NK cells, TGF-β1, IL-18, cytokines, NKp30, activating receptors, CXCR4, chemokine receptors, p38MAPK, MEK/ERK

## Abstract

TGF-β1 is a pleiotropic factor exerting a strong regulatory role in several cell types, including immune cells. In NK cells it profoundly alters the surface expression of crucial activating and chemokine receptors. To understand which soluble signals might better contrast these effects, we cultured human NK cells in the presence of TGF-β1 and different innate and adaptive cytokines, generally referred as “immunostimulatory”. These included IL-2, IL-15, IL-21, IL-27, and IL-18. Unexpectedly, IL-18 strengthened rather than contrasting important TGF-β1-mediated functions. In particular, IL-18 further reduced the expression of CX_3_CR1 and NKp30, leading to the virtual abrogation of the triggering capability of this activating receptor. Moreover, IL-18 further increased the expression of CXCR4. The IL-18-mediated additive effect on NKp30 and CXCR4 expression involved transcriptional regulation and activation of MEK/ERK and/or p38MAPK. A proteomic approach quantified both surface and intracellular proteins significantly modified in cytokine-treated NK cells, thus giving global information on the biological processes involving TGF-β1 and IL-18. Our data support the concept that IL-18 may have a different behavior depending on the type of soluble factors characterizing the microenvironment. In a TGF-β1 rich milieu such as tumors, it may contribute to the impairment of both NK cells recruitment and killing capability.

## 1. Introduction

TGF-β1 is a pleiotropic factor secreted by several cell types in both homeostatic and pathological conditions [1,2]. It plays a crucial role in maintaining peripheral tolerance and regulating the magnitude of immune responses [3]. Mouse models show that TGF-β1 deficiency leads to multi-organ inflammation and early death [4,5], outcomes that are recapitulated by the absence of TGF-β1 activation [6,7]. Indeed, TGF-β1 is secreted in latent/inactive forms [1,3], non covalently associated with the latency-associated peptide (LAP) in small latency complexes (SLC) or, most commonly, in large latency complexes (LLC) consisting of SLC and the latent TGF-β1 binding protein (LTBP). Under physiological conditions large amounts of inactive TGF-β1 are present in body fluids and tissues that, where and when necessary, can be rapidly activated by different mechanisms. LLCs anchor to extracellular matrix components and TGF-β1 can be released via integrins expressed at the cell surface. Additional modalities of TGF-β1 activation include the action of the membrane glycoprotein thrombospondin-1, shear stress, proteases, heat, acidic pH, and reactive oxygen species [3]. 

The free/active form of TGF-β1 exerts a plethora of functions on a variety of cells belonging to both the non hematopoietic and hematopoietic compartment. To mention a few, TGF-β1 drives Epithelial to Mesenchymal Transition (EMT) [8], is responsible for hibernation of hematopoietic stem cells (HSC) in the bone marrow niches [9], and contributes to the polarization of different T cell types [10]. TGF-β1 also regulates the main functions of Natural Killer (NK) cells. It inhibits the IFN-γ production induced by the synergic activity of IL-12, IL-15, and IL-18, cytokines that can be released by macrophages and/or DC [11]. In this regard a reciprocal antagonism exists between these cytokines and TGF-β1. IL-12, IL-15, and IL-18 (although with a different strength) have been show to suppress TGF-β1 signals through the downregulation of TGF-βRII, SMAD2, and SMAD3 proteins [11]. Conversely, TGF-β1 induces SMAD2, 3, and 4 that suppress T-bet and antagonize IFN-γ gene expression negatively regulating the promoter [11]. TGF-β1 is also capable of modifying the surface phenotype of NK cells. In particular, it reduces the surface expression of crucial activating NK receptors (NKp30 and NKG2D), an affect that can impair tumors killing and DC selection [12,13,14]. Moreover, we have recently highlighted the role played by TGF-β1 in modifying the chemokine receptor repertoire of NK cells, with possible impact on NK cell migration [15,16]. In particular, it modulates the surface expression of CXCR3, CXCR4, and CX_3_CR1, and miR27a-5p has been identified as a regulator of the TGF-β1-mediated downregulation of CX_3_CR1 [17]. 

All these observations prompted us to compare different cytokines, generally referred as “immunostimulatory” for their capability of contrasting the modulatory effects of TGF-β1, focusing on its effect on activating and chemokine receptors. Cytokines included IL-18 that, unexpectedly, reinforced rather than weakened some TGF-β1-mediated effects, thus showing not yet described immunomodulatory properties.

## 2. Results

### 2.1. IL-18 Boosts the Effect of TGF-β1

As previously shown [12], optimal concentrations of TGF-β1 induced downregulation of the NKp30 and NKG2D activating receptors, and modified the chemokine receptor repertoire of peripheral blood (PB) NK cells [15,16]. In particular, it increased the expression of CXCR4 and CXCR3 and decreased that of CX_3_CR1 (Figure 1). In search for cytokine(s) that could modify these TGF-β1-mediated immunomodulatory effects, we cultured peripheral blood (PB) NK cells in the presence of TGF-β1 alone or in combination with different cytokines known to exert immunostimulatory function (Figure 1 and Appendix A).

IL-2 and IL-15 significantly contrasted the TGF-β1-mediated downregulation of the activating receptors, which recovered expression levels comparable (or even higher) to untreated NK cells. On the contrary, IL-21, IL-12, IL-27, and IL-32 had no effects (Figure 1 and Appendix A). IL-18 was unable to restore NKG2D expression, despite its ability to upregulate its expression when used alone (Appendix A). Notably, IL-18 showed an unexpected immunomodulatory role, significantly strengthening the TGF-β1-mediated downregulation of NKp30 surface expression (Figure 1).

Regarding the chemokine receptor repertoire, none of the cytokines analyzed influenced the effect of TGF-β1 on CXCR3 expression. IL-2 or IL-15 did not alter the effect of TGF-β1 on CX_3_CR1 while strongly contrasted the CXCR4 upregulation (Figure 1), leading to chemokine receptor surface levels even lower than those present in untreated NK cells. Again, IL-18 showed very peculiar behavior. Indeed, it significantly potentiated the activity of TGF-β1 further reducing the expression of CX_3_CR1 and increasing that of CXCR4 (Figure 1). The latter effect was evident in both CD56^dim^ and CD56^bright^ NK cells (Appendix A). Notably, a significant decrease of CX_3_CR1 surface levels was detected also when IL-18 was used alone (Appendix A). Moreover, IL-18 also decreased the expression of CXCR1, both when used alone (Appendix A) or in the presence of TGF-β1 (Figure 1). Among the other cytokines analyzed, IL-21 significantly opposed the TGF-β1-mediated upregulation of CXCR4, although to a lower extent as compared to IL-2 or IL-15 (Figure 1). On the other hand, IL-21 shared with IL-18 the property of further reducing the CX_3_CR1 surface levels resulting from TGF-β1 conditioning. IL-12, IL-27, or IL-32 did not modify the TGF-β1-mediated modulatory effect on CXCR4 and CX_3_CR1 expression (Figure 1 and Appendix A).

It is of note that the additive effect of IL-18 on TGF-β1-mediated upregulation of CXCR4 was preserved using TGF-β1 concentrations ≥1 ng/mL (Appendix A). On the contrary, that impacting on NKp30 surface levels was lost using suboptimal concentrations of TGF-β1 (Appendix A).

Besides the peculiar immunomodulatory behavior described above, IL-18 shared with IL-15 typical immunostimulatory functions. Indeed, NK cells treated with IL-18 alone showed increased surface levels of the NKG2D receptor as well as of CD69 and CD25 activation markers (Appendix A).

### 2.2. TGF-β1 Plus IL-18 Affect NKp30-Mediated Triggering

We analyzed whether the very low NKp30 surface densities resulting from TGF-β1 plus IL-18 conditioning impacted on the killing capability of NK cells. To this end, PB NK cells were treated with the cytokines combination and analyzed in ADCC against the FcγR+ P815 target cells [18]. As shown in Figure 2A, differently from untreated cells, the mAb-mediated engagement of NKp30 in TGF-β1 plus IL-18 conditioned NK cells was virtually unable to induce lysis of the target. On the contrary, the engagement of CD16 still induced optimal cytotoxicity. This result suggested that, the combined action of these cytokines selectively affected the triggering capability of NKp30 rather than significantly impacting on the overall cytolytic potential of NK cells (i.e., granzyme/perforin cellular content).

In the same set of experiments we analyzed whether classical immunostimulatory cytokines such as IL-2 or IL-15 could oppose the TGF-β1 plus IL-18 immunomodulatory effects. Results showed that both cytokines induced in NK cells the recovery of the NKp30 surface expression (Figure 2B and Appendix A). This was paralleled by restoration of receptor’s function. Indeed, mAb-mediated engagement of NKp30 resulted in a strong cytolytic activity against target cells, significantly higher as compared to untreated PB-NK cells (Figure 2A).

IL-2 and IL-15 also abrogated the CXCR4 upregulation induced by TGF-β1 plus IL18 (Figure 2B and Appendix A).

### 2.3. Molecular Mechanisms Involved in the Additive Effect of IL-18

We analyzed whether transcriptional regulation was involved in the IL-18-mediated additive effect on TGF-β1 activity. PB NK cells were treated with different concentrations of TGF-β1 in the absence or in the presence of IL-18 (or IL-15) and analyzed by quantitative PCR for CXCR4 and NKp30 mRNA expression. qPCR for CD25 mRNA was also performed as control. As shown in Figure 3, treatment with optimal doses of TGF-β1 resulted in an increase of CXCR4 mRNA and reduction of NKp30 mRNA, effects that were significantly potentiated by IL-18. The addition of IL-18 to suboptimal doses of TGF-β1 (2.5 ng/mL) did not cause any significant change in NKp30 mRNA levels, according to results on receptor expression depicted in Appendix A. On the contrary, although significant upregulation of CXCR4 at protein level was detected upon addition of IL-18 to suboptimal doses of TGF-β1 (both at 2.5 and 1 ng/mL) (Appendix A), this effect did not correlate with increased mRNA levels, suggesting the release to the cell surface of preformed cytoplasmic CXCR4 (Figure 3). As expected, IL-18 alone caused a significant upregulation of CD25 mRNA, an effect shared by IL-15. IL-15 also contrasted the effect of TGF-β1, significantly increasing NKp30 and reducing CXCR4 mRNA levels (Figure 3).

p38 MAPK and MEK/ERKs are involved in the transduction pathway of IL-18 and TGF-β1 [19,20]. To analyze their possible involvement in the additive effect of IL-18 on TGF-β1 activity, PB NK cells were treated with these cytokines in the absence or in the presence of specific kinase inhibitors, and analyzed by flow cytometry for NKp30 and CXCR4 expression (Figure 4 and Appendix A). The capability of TGF-β1 alone to modulate NKp30 and CXCR4 expression was not significantly affected either by p38 MAPK or MEK/ERKs inhibitors, suggesting a predominant intervention of the SMAD signaling pathway. Differently, p38 MAPK and MEK/ERKs were involved in the additive effect of IL-18 on TGF-β1. In particular, the p38 MAPK inhibitor abolished (NKp30) or significantly reduced (CXCR4) the additive effect of IL-18 (Figure 4 and Appendix A). The NKp30 downregulation was also significantly affected by inhibition of MEK/ERKs. On the contrary, MEK/ERKs inhibition did not cause significant variation in upregulation of CXCR4 (Figure 4 and Appendix A).

### 2.4. Effect of IL-18 Plus TGF-β1 on NK Cell Proteome

To gain insight on the effects of TGF-β1 and IL-18 on NK cell phenotype and function we used a proteomic approach. PB-NK cells untreated (CTR) or conditioned for 48 h with TGF-β1, alone (TGF-β1 data set), or in combination with IL-18 (TGF-β1 plus IL-18 data set) were lysed and analyzed by high-resolution mass spectrometry. Data processing through the MaxQuant software allowed the identification of a total of 4331 proteins (of which 3971 were quantified using a Label-Free Quantitation approach).

A bioinformatics enrichment analysis of GO functional terms revealed biological categories harboring proteins that were significantly modified by TGF-β1 vs. CTR, TGF-β1 plus IL-18 vs. CTR or TGF-β1 data sets (Figure 5). Diverging flow lines showed biological activities modulated by TGF-β1 or IL-18, which included the known involvement of IL-18 in the IFN-γ-mediated signaling pathway and antigen processing/presentation [21,22,23]. Some converging biologic processes were identified that were highly enriched in significantly modulated proteins, as indicated by width of the flow lines. These included exocytosis, antimicrobial immune responses and cell-to-cell adhesion in the TGF-β1 plus IL-18 vs. CTR data set, or viral transcription/infectious cycle and intracellular protein targeting in the TGF-β1 plus IL-18 vs. TGF-β1 data set.

Next two volcano plots were generated from TGF-β1 vs. CTR and TGF-β1 plus IL-18 vs. TGF-β1 data sets, and representative proteins selected on the basis of a two-sample *t*-test (FDR = 0.05 and s0 = 0.1 log10 of *p*-value, *y*-axis) (Figure 6A,B). Among these, 40 proteins were regulated by both TGF-β1 and IL-18 with Student’s *t*-test difference or –log Student’s *t*-test < or >1. Focusing on proteins expressed at the cell surface membrane, few molecules emerged from our selection and, differently from cytofluorimetric results, in no cases an additive effect of IL-18 was observed. In particular, the protein level of CXCR4 was significantly increased by TGF-β1 (Figure 6A) but the additive effect of IL-18 was not appreciated (Figure 6B). Moreover, no decrements in the protein levels of NKp30 (or NKG2D) were detected in the presence of TGF-β1 used alone or in combination with IL-18.

Proteomic quantified both surface and intracellular proteins, thus giving global information on the cellular state. However, the results showed that sensitivity of this approach was considerably lower as compared to flow cytometry. Nonetheless, looking at opposite effects of TGF-β1 and IL-18, we detected a strong modulation of the C-Type Lectin Domain Containing protein 14A (CLEC14A), a surface molecule involved in cell-to-cell contact and angiogenesis. As shown in Figure 6A, CLEC14A protein level strongly decreased in the presence of TGF-β1, an effect that was deeply contrasted by IL-18 (Figure 6B). Interestingly CLEC14A [24,25] has been never described in NK cells and no data on its expression and function are available so far. This and other proteins (among the 40 selected) showing co-regulation by IL-18 and TGF-β1 will be subjected to further investigations.

## 3. Discussion

TGF-β1 has been shown to exert a huge number of immunomodulatory functions some of which play a detrimental role in cancer patients.

Focusing on NK cells it inhibits the activation of the kinase mTOR, a central regulator of NK cell metabolism and function [26]. Moreover TGF-β1 profoundly alters the surface expression of crucial activating and chemokine NK receptors. To understand which soluble signals might better contrast these effects, we treated peripheral blood NK cells with TGF-β1 and different immunostimulatory cytokines.

This analysis highlighted an unpredictable behavior of IL-18 that, differently from other typical immunostimulatory cytokines such as IL-2 and IL-15, strengthened rather than weakening some TGF-β1-mediated effects. In particular, IL-18 potentiated the TGF-β1-mediated downregulation of NKp30, with a virtual total loss of its capability to trigger the NK cells cytolytic activity. This might lead to a decreased ability of NK cells to kill targets that express the specific ligand(s), which include tumors of different histotype [27] and tumor-associated endothelial cells [28]. Notably, the NKp30 receptor is also involved, together with DNAM-1, in NK/DC interactions that lead to reciprocal cell activation [29,30,31,32]. DCs undergoing an appropriate maturation program are spared from NK-cell mediated attack since they express very high levels of MHC class I molecules. On the contrary, DCs undergoing an unfruitful maturation process, characterized by an inadequate MHC class I upregulation, are killed and the NKp30 receptor play a crucial role in this process. The combined action of TGF-β1 and IL-18 on NKp30 expression may significantly impair the NK-mediated tumor immune surveillance as well as the “DC editing” process, allowing the survival of MHC class I^low^ DCs that, due to the inappropriate antigen presentation, could promote tolerogenic responses.

Our data suggest that TGF-β1 and IL-18 may also profoundly modify the migratory capability of NK cells, limiting their extravasation and their encounter with other cell types present in inflamed tissues including tumor cells. Indeed, TGF-β1 and IL-18 alter the expression of chemokine receptors that play a key role in NK cells homing/egress to/from bone marrow (BM), interaction with endothelium, and recruitment into peripheral tissues [33,34,35,36]. In particular, they reduce the expression of CX_3_CR1, and IL-18 also decreased the expression of CXCR1. These chemokine receptors are selectively expressed by CD56^dim^ NK cells and regulate their migration toward CX_3_CL1 (fractalkine) and CXCL8, respectively [16]. CD56^dim^ NK cells represent the majority of cells circulating in blood, and are highly cytolytic (Perforine^high^) CD16^pos^ effectors, fully responsive to the engagement of activating receptors such as NKp30 [37]. Thus, it is conceivable that CX_3_CR1^low^ CXCR1^low^ CD56^dim^ cells may show defective migration toward tumor or inflamed tissues. In this context, a low expression of CX_3_CR1 has been reported in CD56^dim^ NK cells in both peripheral blood and tumor-infiltrated bone marrow (BM) of neuroblastoma patients [15].

TGF-β1 and IL-18 also dramatically increased the expression of CXCR4, a process that involves p38MAPKactivation, increased transcription and, possibly, protein mobilization from the intracellular stores. This modulatory effect may favor the migration in tissues of poorly cytolytic (perforine^low^) CXCR4^pos^, CXCR3^pos^, CD56^brigh^ immature NK cells, and/or the retention in the BM of CXCR4^pos^ immature and mature NK cells thus reducing the pool of circulating NK cells committed to tumor surveillance. Along this line it should be interesting to analyze whether IL-18 might cooperate with TGF-β1 in the reduction of NK cells number and/or functions, favoring the differentiation of non cytolytic members of the Innate Lymphoid cells family. Indeed it has been shown that TGF-β1 negatively impacts on differentiation of NK cells [38], leading to the onset of peripheral blood cells with a decidual NK-like phenotype. TGF-β1 is also capable of converting differentiated human CD16+ NK cells in poor cytolytic CD16- decidual-like NK cells. Moreover different mouse models have highlighted novel crucial regulatory functions for TGF-β1. In particular in mice in which ILC and NK cells lack SMAD4, a noncanonical TGF-β1-mediated signal drives conversion of NK cells into ILC1 [39]. TGF-β1 has been also described to promote NK cell conversion in intermediate ILC1 cells unable, like ILC1, to control tumor metastasis. Moreover, by an autocrine loop TGF-β1 sustains the expansion of ILCreg, a recently identified subpopulation that colonizes mouse and human intestines producing IL-10 [40], a cytokine known to negatively regulate immune system.

A major source of IL-18 is represented by macrophages [41,42,43] which can also release and activate TGF-β1 [44]. During inflammation, the role of IL-18 can change depending on the predominant macrophage polarization. In acute inflammation such as in response to pathogen-related stimuli, IL-18 produced by M1-polarizing macrophages may synergize with IL-12 and IL-15, thus exerting an immunostimulatory rather that immunoregulatory role. In chronic inflammation processes such as those occurring in tumor tissues, IL-18 could support the function of TGF-β1 that is produced/activated by M2-polarized tumor-associated macrophages (TAM). In this context, a membrane-bound form of IL-18 (mIL-18) has been described that is expressed by a subset (30–40%) of unpolarized (M0) and M2-polarized macrophages, and by the majority of TAM [42,45]. The mechanisms responsible for IL-18 membrane retention and release still remain enigmatic. Interestingly, IL-18 shows many predictable cleavage sites of MMP-2 and -9 (Protease specificity prediction server, PROSPER), extracellular proteases produced by different cell types including M2/TAM [46], which are also involved in TGF-β1 activation. Considering that also tumor cells may represent a source of MMPs, TGF-β1, and IL-18, the combined effect of TGF-β1 and IL-18, supported by the proteases intervention, might play a pivotal role during M2 polarized immune responses occurring in the tumor microenvironment. It should be mentioned that in addition to cleavage by MMP, IL-18 release also requires the inflammasome pathway. In this context, one preclinical study on multiple myeloma demonstrated that the inflammasome-derived IL-18 critically contributes to disease progression driving the generation of myeloid-derived suppressor cells (MDSCs) [47]. According to a possible negative impact on disease progression of tumor microenvironment-derived IL-18, in several tumor types high concentrations of IL-18 in blood or bone marrow correlated with advanced tumor stages and were significantly, and independently associated with shorter overall survival [47,48]. Moreover, IL-18 has been shown to upregulate, on NK cells, PD-1 receptor, capable of delivering an inhibitory signal upon interaction with its PD-Ls ligands [48,49,50].

## 4. Materials and Methods

### 4.1. Cells Used in the Study

NK cells were purified by “Human NK Cell Isolation kit” (Miltenyi Biotec, GmbH, Bergisch Gladbach, Germany) from PBMC of healthy volunteer blood donors admitted at the blood transfusion center of IRCCS S. Martino-IST after obtaining informed consent and the study was approved by the Ethics Committee of IRCCS S. Martino-IST (39/2012). The degree of purity of the isolated NK cells (CD3-, CD56+, and NKp46+) was >98%.

### 4.2. Monoclonal Antibodies and Cytokines

The following mAbs were produced in our laboratory: AZ20 (IgG1) (anti-NKp30), BAT221 (IgG1) (anti-NKG2D), MAR93 (IgG1) (anti CD25), and C227 (IgG1) (anti CD69). Anti-human CXCR4 (IgG2b) and anti-human CXCR3 (IgG1) were purchased from R&D Systems (Minneapolis, MN, USA). Anti-human CXCR1 (IgG1) mAb was purchased from Santa Cruz Biotechnology (Santa Cruz, CA, USA). Anti-human CX_3_CR1 PE (rat IgG2b) and the isotype control (rat IgG2b-PE) were purchased from MBL international (Woburn, MA, USA).

Human recombinant cytokines were purchased from PeproTech (TGF-β1, IL-15, IL-12, and IL-21), Novus Biologicals (IL-32), Proleukin (IL-2), MBL International (IL-18), and R&D Systems (IL-27). NK cells were cultured for 48 h in the presence of RPMI 1640 completed medium (supplemented with 2 mM glutamine, 50 mg/mL penicillin, 50 mg/mL streptomycin, and 5% heat-inactivated FCS) supplemented with the different cytokines used at the following published working concentrations; IL-12: 1 ng/mL; IL-15: 20 ng/mL; IL-18: 100 ng/mL; IL-21: 20 ng/mL; IL-27: 100 ng/mL; and IL-32: 50 ng/mL.

### 4.3. Flow Cytometry

For cytofluorimetric analysis (FACSCalibur Becton Dickinson & Co, Mountain View, CA, USA) cells were stained with the appropriate PE conjugated mAbs or with unconjugated mAbs followed by PE-conjugated isotype-specific goat anti-mouse second reagent (Southern Biotechnology Associated, Birmingham, AL). Isotype-matched irrelevant mAbs were used as control. On every experimental session, the flow cytometer performances were monitored and the reproducibility of the fluorescence intensity was aligned by calibrated microsfere (Becton Dickinson & Co, Mountain View, CA, USA).

### 4.4. Real-Time PCR

Total RNA was extracted from NK cells using the miRCURY RNA Isolation Kit—Cell and Plant (Exiqon)—according to the manufacturer guidelines. Two-hundred-and-fifty nanograms of RNA were reverse transcribed using the SuperScript VILO cDNA Synthesis Kit (Invitrogen, Carlsbad, CA, USA). Real-time PCR was performed using specific TaqMan Gene Expression Assays (Applied Biosystems, Foster City, CA, USA). CXCR4, NKp30 and CD25 gene expression was normalized to glyceraldehyde 3-phosphate dehydrogenase (GAPDH) gene expression. Experiments were performed in triplicate.

### 4.5. p38MAPK and MEK/ERK Inhibitors

Inhibitor SB203580 (p38MAPK) and PD98059 (MEK/ERK) were purchased from Selleckchem. Following recommended procedures both inhibitors were diluted in DMSO (final concentration 0.1%) and in complete RPMI medium in order to obtain the best working concentrations (20 μM for SB203580 and 80 μM for PD98059). Resting NK cells (2 × 10^5^) were then incubated with p38MAPK inhibitor or MEK/ERK for 1 h or 72 h respectively or with DMSO (0.1%) as control. After two washing cycles in complete RPMI medium NK cells were seeded in 96 round bottom wells and conditioned by cytokines for 2 days

### 4.6. Mass Spectrometry Based Proteomics

The samples have been processed by iST protocol [51].Each digested sample was analyzed by high-resolution liquid chromatography and tandem mass spectrometry (LC–MS/MS) based on Orbitrap technology. The quantification strategy is a label free approach (LFQ) available in MaxQuant suite [52].

The proteomics data are subjected to a statistical validation applying tools developed in Perseus Software [53]. Raw data will be available upon request. Additional methodological information is available as Appendix A.

### 4.7. Statistical Analysis

Statistical analysis with level of significance (*p*) and graphic representation were performed using Wilcoxon–Mann–Whitney *p*-value test (nonparametric significance test) and GraphPad Prism 6 (GraphPad Software La Jolla, CA, USA).

## 5. Conclusions

The present study describes an additive effect of IL-18 on some crucial regulatory functions of TGF-β1, strongly supporting the concept that IL-18 must be considered “more than a Th1 cytokine” [54]. Initially identified as an interferon (IFN)-γ inducing factor capable of inducing type 1 responses, IL-18 may exert opposite effects by supporting cytokines with either immunostimulatory (IL-12, IL-15, or IL-2) or immunosuppressive (TGF-β1) functions. 

The different IL-18 behavior might depend on the type, the relative amount and the timing release of the soluble factors characterizing the microenvironment. In a TGF-β1-riched milieu such as tumors IL-18 might play a tumor-promoting role decreasing the function of NK cells and reducing the migration of highly cytolytic CD56^dim^ NK cells. 

Increased local or systemic IL-18 levels were detected in different Th2-mediated disorders [55] and very recently NK cells have been shown to kill Th2-polarizing DCs via the engagement of NKp30 [56]. Thus the ability of IL-18 and TGF-β1 to negatively regulate NKp30 expression might also enrich the pool of Th2-polarizing DCs, promoting type 2 immune responses. 

To conclude our data reveal novel unexpected functions of IL-18 and pave the way for further studies. In particular it should be relevant to address whether IL-18 also supports the ability of TGF-β1 to hamper the differentiation of NK cells while favoring that of non-cytolytic members of the innate lymphoid cells family.

## Figures and Tables

**Figure 1 cancers-11-00075-f001:**
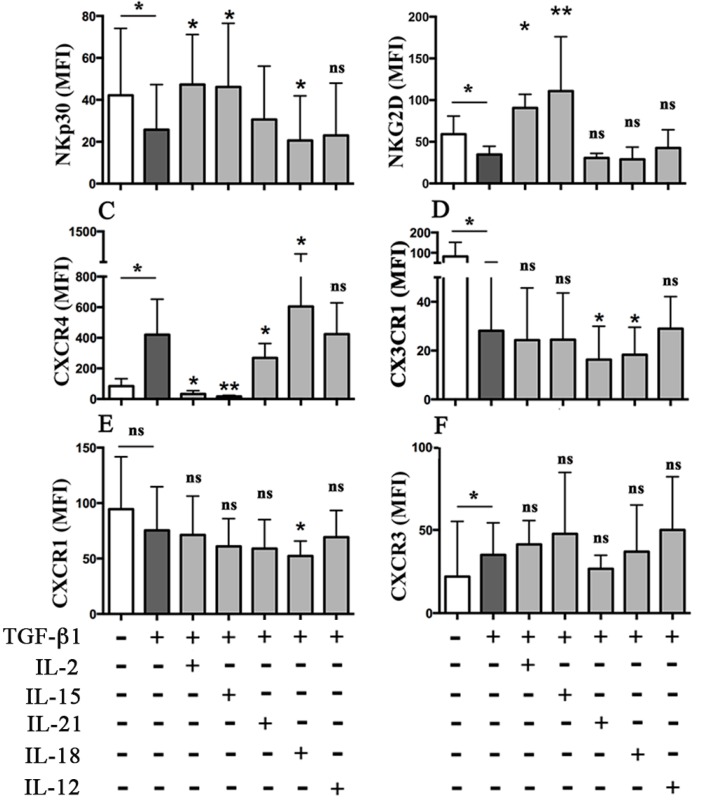
Effect of pro-inflammatory cytokines on the main TGF-β1-mediated immunomodulatory effects. NK cells purified from peripheral blood of healthy donors (PB NK) were cultured for 48 h either in the absence or in the presence of TGF-β1 used alone or in combination with the indicated cytokines. Cells were analyzed by flow cytometry for the expression of activating and chemokine receptors. Average of six independent experiments (six unrelated healthy donors). Mean fluorescence intensity (MFI) and 95% confidence intervals are shown. ns: not significant, * *p* < 0.05, ** *p* < 0.01. If not indicated, statistical significance is referred to results obtained with TGF-β1 alone.

**Figure 2 cancers-11-00075-f002:**
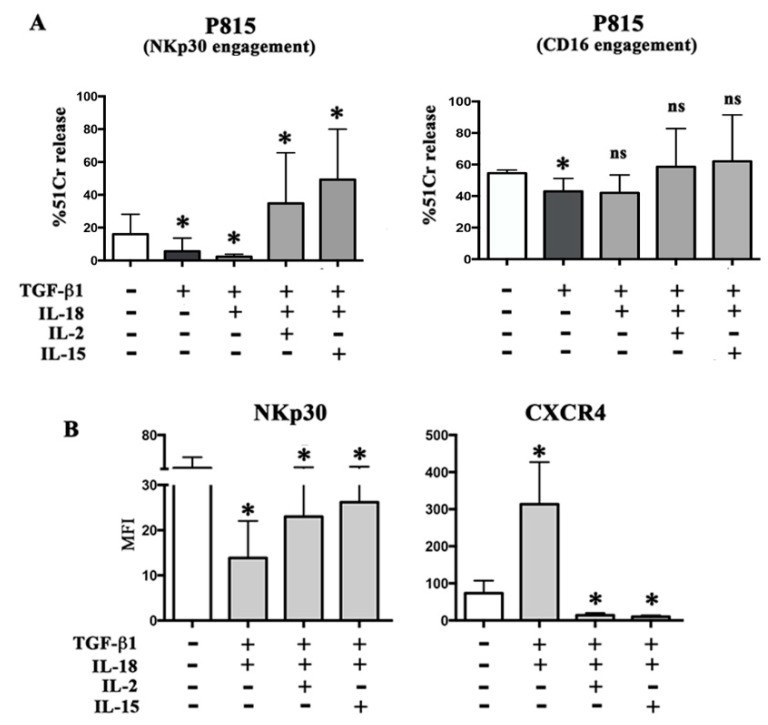
IL-2 or IL-15 restores receptor expression and function. PB NK, cultured for 48 h, in the absence or in the presence of the indicated cytokine combinations, were analyzed for cytolytic activity against the FcγR+ P815 target cell in the presence of mAb specific for NKp30 or CD16 (^51^Cr release) (panel **A**), and for the expression of NKp30 and CXCR4 (flow cytometry) (panel **B**). Average of four independent experiments (four donors). Mean of MFI and 95% confidence intervals are shown. * *p* < 0.05; ns means *p* not significant. Statistical significance has been referred to cells cultured without cytokines (white bar).

**Figure 3 cancers-11-00075-f003:**
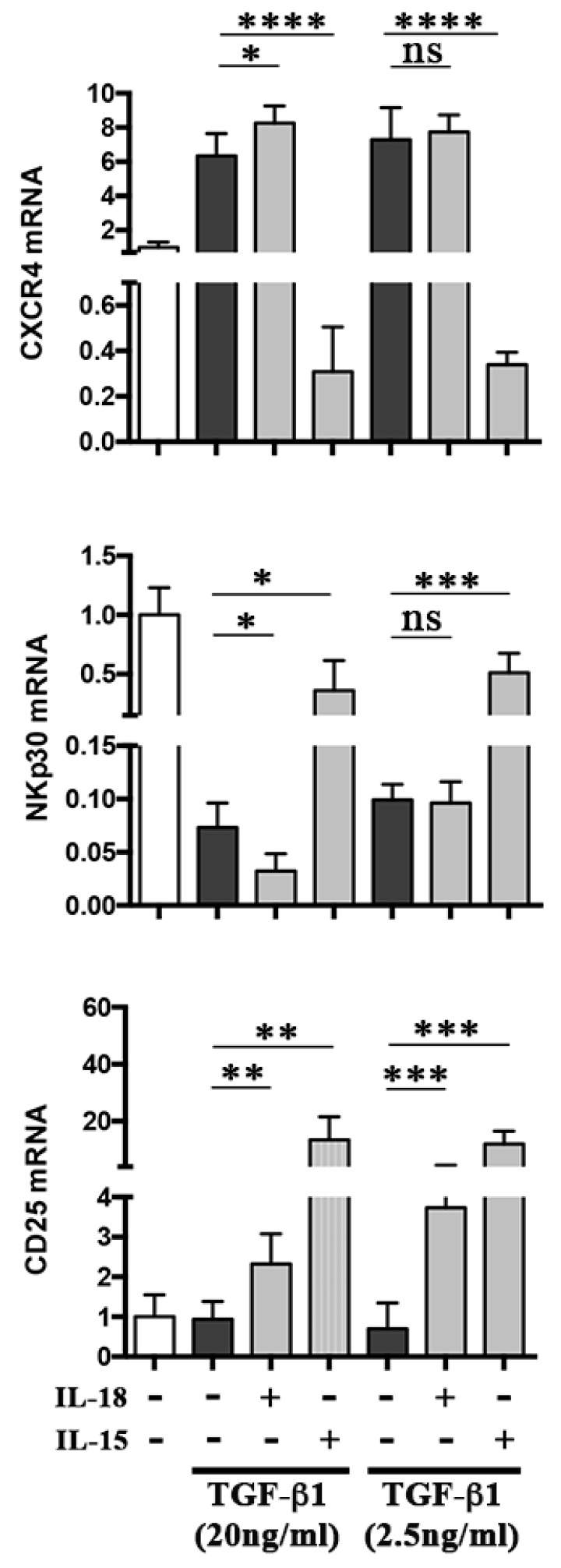
CXCR4 and NKp30 mRNA expression in NK cells treated with TGF-β1 plus IL-18. PB NK cells untreated or treated for 48 h with the indicated amounts of TGF-β1, alone or in combination with IL-18 or IL-15, were analyzed by qPCR for CXCR4, NKp30, and CD25 mRNA expression. Average of five independent experiments (five donors, each analyzed in triplicate). Glyceraldehyde 3-phosphate dehydrogenase (GAPDH) has been used as reference control. ns: not significant. * *p* < 0.05, ** *p* < 0.01, *** *p* < 0.001, **** *p* < 0.0001.

**Figure 4 cancers-11-00075-f004:**
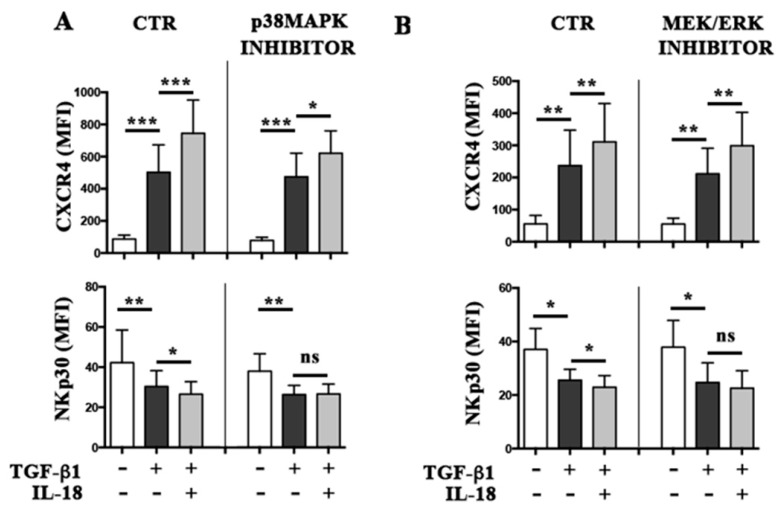
Involvement of p38MAPK or MEK/ERK in the additive effects mediated by IL-18 on TGF-β1 activity. Cytofluorimetric analysis of CXCR4 and NKp30 surface expression in PB NK cells treated with TGF-β1 or TGF-β1 plus IL-18, in the absence or presence of p38MAPK (panel **A**) or MEK/ERK (panel **B**) inhibitors. Average of four independent experiments (four donors). Mean of MFI and 95% confidence intervals are shown. ns: not significant, * *p* < 0.05, ** *p* < 0.01, *** *p* < 0.001.

**Figure 5 cancers-11-00075-f005:**
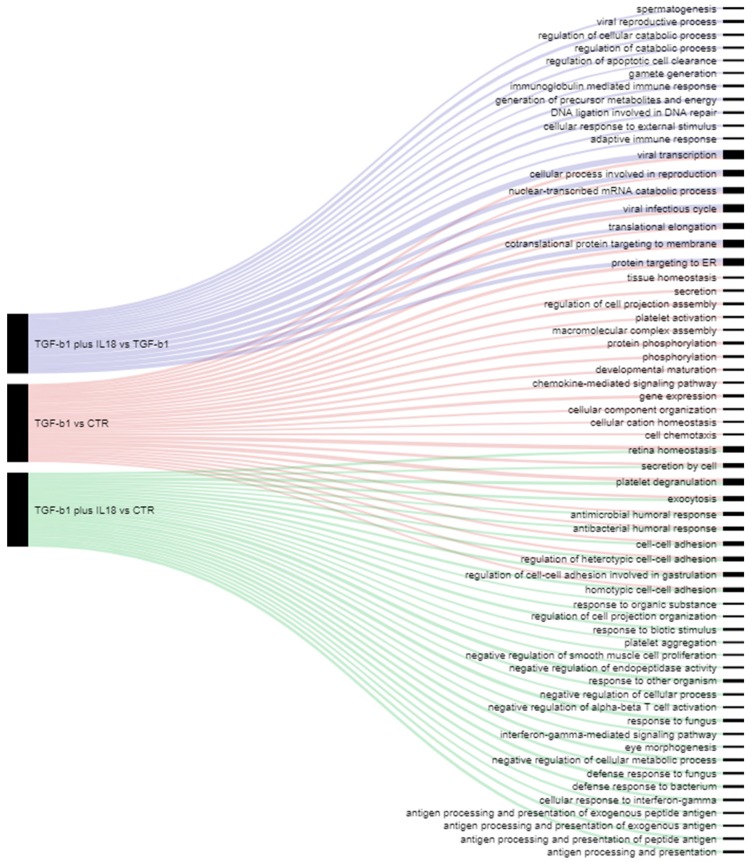
Profile plot of 1D annotation enrichment results. Enrichment is done by GOBP downloaded from Uniprot with a significance threshold of *p* < 0.001. The output file is converted into a Sankey plot where different colors are associated with different data sets and the width of flow line for each biological process is related to −log10 (*p*-value) of the enrichment.

**Figure 6 cancers-11-00075-f006:**
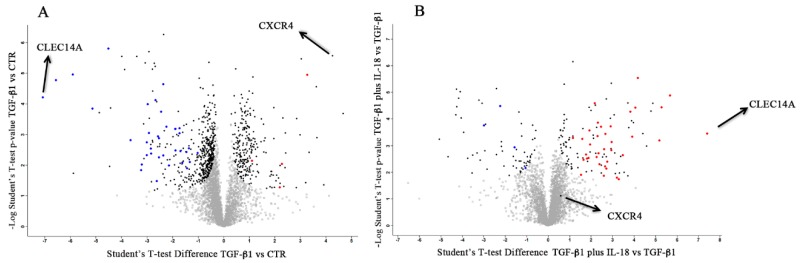
Proteomic analysis of NK cells unconditioned or conditioned with TGF-β1 (vs. CTR) or TGF-β1 plus IL18 (vs. TGF-β1). Volcano plot representations of differentially expressed proteins relative to TGF-β1 vs. CTR (unstimulated) (**A**) or TGF-β1 plus IL-18 vs. TGF-β1 data sets. (**B**) Black dots represent proteins that display both large magnitude fold-changes (*x*-axis, proteins upregulated after treatment are shown on the right) as well as high statistical significance (FDR = 0.05 and s0 = 0.1 log10 of P value, *y*-axis). Colored plots represent 40 proteins more significantly downregulated (blue) or upregulated by TGF-β1 treatment (vs. CTR) o by TGF-β1 plus IL-18 (vs. TGF-β1). Gray squares represent proteins not significantly modulated by the treatments. CXCR4 and CLEC14A proteins are indicated in both volcano plots.

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
