# Peer review of "Novel Immunoregulatory Functions of IL-18, an Accomplice of TGF-β1"

_cancers, 2019, doi:10.3390/cancers11010075_

Round 1
Reviewer 1 Report
Casu et al have demonstrated that, unlike other cytokines such as IL2 and IL15, the cytokine IL18 either acts in concert with or does not oppose the function of TGF-b on NK cell chemokine receptor expression of CXCR4 or NKp30 expression. The authors also examine how these cytokines affects the cytolytic and signaling activity of human NK cells. The authors additionally use a proteomics approach to look at how these cytokines may regulate protein expression in NK cells.
Major:
There are instances where the authors overstate the findings in the text. While the change may be statistically significant, at times the fold change itself is smaller and it may be more accurate to state that IL18 is not changing TGF-beta activity. IL18 does not always appear to increase the magnitude of the TGF-beta effect on the NK cells.
Minor:
Overall the paper is well written but needs improvement in the English. Specifically, the authors often say that a cytokine “contrasts” or “contradicts” the activity, but it would be more correct to say that the cytokine “mitigates” or “opposes” the activity.
In Figure 2A, it would be helpful to see the activity of IL18 or TGF-beta alone so that it is clear how big the synergistic effect is from each cytokine.
On line 189, the last comparison should be more explicitly listed, ie instead of “TGF-b1 sets” it should be TGF-b1 vs TGFb1 plus IL18.
It would be helpful for the label in Figure 5 to show “vs” to indicate the specific comparison. Currently, the underscore does not make the comparison clear since it is used in place of a space bar and is not easily readable.
Author Response
Major:
We agree with the Reviewer that some effects of IL-18 on TGF-b1 activity are smaller than others. However, only statistically significant data have been taken into consideration and, with the exception of qPCR and proteomic analyses, data are represented as MFI (flow cytometry) or % 51Cr release (cytotoxicity) and not as fold changes. It should be also considered that the modulatory effect of TGF-b1 alone is really potent. In spite of this we appreciated further modulation with IL-18, effects that were not observed with other cytokines used for comparison.
Minor:
1) As suggested “ to contrast” or “to contradict” have been replaced with “to oppose”.
2) As suggested, we added in Figure 2A data showing the activity of TGF-b1 alone.
3) As suggested, “TGF-b1 sets” has been replaced with “TGF-b1 vs TGFb1 plus IL-18” on line 189
4) As suggested we used “vs” to clarify the date sets comparison showed in both Figure 5 and 6.
Changes are tracked in the submitted revised version.
Reviewer 2 Report
In this manuscript entitled “Novel immunoregulatory functions of IL-18, an accomplice of TGF-β1”, Casu et al. showed that IL-18 negatively regulate NK cell function in the presence of TGF-beta. IL-18 was originally identified as an interferon-gamma inducing factor, and has been recognized as an anti-tumor cytokine that stimulates NK cells. However, growing evidence suggest that IL-18 also has immunoregulatory functions. Using human NK cells, this study showed that IL-18 downregulates Nkp30 expression and CX3CR1 levels in the presence of TGF-beta, suggesting that IL-18 could negatively regulate NK cell function in the tumor microenvironment. Overall, this study has interesting/novel findings, and data was supported by proteomics analysis.
In my view, one major concern is Discussion part. This part should be re-organized. Authors describe broad aspects/functions of IL-18/TGFbeta in the context of BM niche (Line 286), macrophages polarization (Line 295), and Th2-related diseases (Line 313). Given that this is submitted to “’Cancers”, discussion should be mainly focused on the possible role of IL-18/TGFbeta in the tumor microenvironment. Instead of broad description of IL-18/TGFb functions, following three points should be discussed with references.
1) Recently, several works showed that non-canonical TGFbeta signalling drives conversion of NK cells into ILC1, and that ILC1-like cells have pro-tumor functions (Cortez et al Nature Immunology 2017 and Gao et al. Nature Immunology 2017). Given that TGF-beta is known to convert human CD16+ NK cells into decidual NK cells (Keskin et al. PNAS 2007), TGFbeta + IL-18 might contribute to generation of ILC1-like NK cells with pro-tumor functions (e.g. proangiogenic NK cells). This point needs to be discussed.
2) Pro-tumor functions of IL-18 need to be described more. For example, IL-18 upregulates PD-1 expression on NK cells (Terme et al. Cancer Res 2011). Indeed, Park et al. showed tumor-derived IL-18 increased PD-1 expression on dysfunctional NK cells in the breast cancer patients (Park et al. Oncotarget. 2017). These previous studies support the negative impact of IL-18 on NK cells.
3) In Discussion part (Line 295-), authors should mentioned that IL-18 is regulated by the inflammasome pathway, in addition to cleavage by MMP. Indeed, one study demonstrated that the inflammasome-derived IL-18 critically contributes to disease progression in multiple myeloma, and that BM IL-18 high patients showed poor prognosis, compared to low patients (Nakamura et al. Cancer Cell 2018). This work supports the negative impact of tumor microenvironment-derived IL-18 on disease progression. These points should be discussed.
Author Response
We thank the reviewer for all comments. The discussion part has been re-organized and implemented according to all the reviewer’s suggestions. References have been cited.
Changes are tracked in the submitted revised version.